# CysLT_2_R Antagonist HAMI 3379 Ameliorates Post-Stroke Depression through NLRP3 Inflammasome/Pyroptosis Pathway in Gerbils

**DOI:** 10.3390/brainsci12080976

**Published:** 2022-07-24

**Authors:** Li Zhou, Jiajia Zhang, Xue Han, Jie Fang, Shasang Zhou, Lingqun Lu, Qiaojuan Shi, Huazhong Ying

**Affiliations:** 1School of Pharmaceutical Sciences, Zhejiang Chinese Medical University, Hangzhou 310053, China; zl18102191690@163.com; 2Zhejiang Provincial Key Laboratory of Laboratory Animals and Safety Research, Hangzhou Medical College, Hangzhou 310013, China; jia9802@163.com (J.Z.); 13819113623@163.com (X.H.); fangjie3318@163.com (J.F.); zhoushasang@163.com (S.Z.); lulingqun@sohu.com (L.L.); shiqiaojuan@163.com (Q.S.)

**Keywords:** cysteinyl leukotriene receptor 2, HAMI 3379, post-stroke depression, neuroinflammation, NLRP3 inflammasome, pyroptosis

## Abstract

Post-stroke depression (PSD) is a kind of prevalent emotional disorder following stroke that usually results in slow functional recovery and even increased mortality. We had reported that the cysteinyl leukotriene receptor 2 (CysLT_2_R) antagonist HAMI3379 (HM3379) contributes to the improvement of neurological injury. The present study was designed to investigate the role of HM3379 in PSD-induced chronic neuroinflammation and related mechanisms in gerbils. The gerbils were subjected to transient global cerebral ischemia (tGCI) and spatial restraint stress to induce the PSD model. They were randomized to receive the vehicle or HM3379 (0.1 mg/kg, *i.p.*) for a consecutive 14 days. In the PSD-treated gerbils, HM3379 had noteworthy efficacy in improving the modified neurological severity score (mNSS) and depression-like behaviors, including the sucrose preference test and the forced swim test. HM3379 administration significantly mitigated neuron loss, lessened TUNEL-positive neurons, and reduced the activation of microglia in the cerebral cortex. Importantly, HM3379 downregulated protein expressions of the NOD-like receptor pyrin domain containing 3 (NLRP3) inflammasome and pyroptosis including NLRP3, cleaved caspase-1, interleukin-1β (IL-1β), IL-18, cleaved gasdermin-N domain (GSDMD-N), and apoptosis-associated speck-like protein containing a caspase activation and recruitment domain (ASC). Mechanistically, HM3379 could repress pyroptosis via inhibiting NLRP3 inflammasome activation under oxygen-glucose deprivation (OGD) stimulation. Knockdown of CysLT_2_R by short hairpin RNA (shRNA) or overexpression of CysLT_2_R by lentivirus (LV)-CysLT_2_R could abolish or restore the anti-depression effect of HM3379. Our results demonstrated that the selective CysLT_2_R antagonist HM3379 has beneficial effects on PSD, partially by suppressing the NLRP3 inflammasome/pyroptosis pathway.

## 1. Introduction

Post-stroke depression (PSD) is one of the most common psychiatric complications in the function recovery of stroke survivors and even increases the risk of recurrent stroke and mortality [1,2]. The prevalence rate of PSD among stroke patients has been estimated to be = up to 33%, with a peak prevalence in the first year [3]. This disease is distinguished by raised cognitive deficits, insomnia, social dysfunction, and hopeless feelings [4,5]. Despite the serious impact of PSD on the quality of life of patients, it is often neglected and untreated.

Cysteinyl leukotrienes (CysLTs), as members of the eicosanoid inflammatory lipid mediator family, are mediators of arachidonic acid-derived lipids [6,7]. They are involved in numerous inflammatory diseases by activating cysteinyl leukotriene receptor 1 (CysLT_1_R) and cysteinyl leukotriene receptor 2 (CysLT_2_R) [8]. Pharmacological studies have shown that CysLT_1_R antagonists, including pranlukast and montelukast, are extensively applied for asthma and allergic rhinitis [9,10]. CysLT_2_R has been reported to be abundantly expressed in the brain, so its role in the pathogenesis of neuropsychiatric disorders may be anticipated [9,10,11]. HAMI 3379 (HM3379), has been known as the first potent and selective CysLT_2_R antagonist [8]. Accumulating evidence, including from our studies, have demonstrated that HM3379 contributes to the improvement of neurological damage in middle cerebral artery occlusion (MCAO), as seen in MCAO-induced rodent models, and to the attenuation of oxygen-glucose deprivation (OGD)-induced neuronal cell injury in vitro [7,8,12,13]. However, it remains unknown whether HM3379 can ameliorate PSD-induced neuronal injury and depression-like behavior in Mongolian gerbils.

Emerging studies have shown that the pathogenesis of PSD is closely associated with multiple inflammatory cascade processes [14,15]. Of all the inflammasome proteins, the NOD-like receptor pyrin domain containing 3 (NLRP3) inflammasome is the most studied. It consists of NLRP3, apoptosis-associated speck-like protein containing a caspase-1 recruitment domain (ASC) and procaspase-1 [16,17,18]. Once NLRP3 is activated, it can induce the maturation and secretion of interleukin-1β (IL-1β) and IL-18, followed by pyroptosis [15,17,18,19,20]. Pyroptosis, as a proinflammatory form of cell death, has been demonstrated to aggravate neuroinflammation following ischemic injury [15,16]. Particularly, microglia can express NLRP3, which results in the secretion of proinflammatory cytokines under cerebral ischemia [21]. Nevertheless, few studies have explored whether the NLRP3 inflammasome/pyroptosis pathway is associated with PSD progression. 

To provide a novel entry point for effective PSD treatment, we investigated the anti-chronic-neuroinflammation effect of HM3379 by inducing PSD in Mongolian gerbils through transient global cerebral ischemia (tGCI) with spatial restraint stress. We hypothesized that this protection effect will be achieved by blocking the NLRP3 inflammasome/pyroptosis pathway. 

## 2. Materials and Methods

### 2.1. Animals

Six-month-old male Mongolian gerbils (weight 70–80 g) were supplied by the Laboratory Animal Center of Hangzhou Medical College (Hangzhou, China). Animals were maintained at 22–26 °C and at 45–65% humidity in a 12 h light–dark cycle and given unlimited food and water. All the experimental procedures were carried out according to the National Institutes of Health Guidelines for the Care and Use of Laboratory Animals. 

### 2.2. Experimental Protocols

Gerbils were randomly divided into three groups: the sham group (sham, n = 6), the tGCI group (n = 6), and the tGCI with HM3379 group (tGCI + HM3379, n = 6). HM3379 (cat# 10580, Cayman Chemical Corporation, Ann Arbor, MI, USA) was dissolved in absolute ethylalcohol and then freshly diluted to 40 μg/mL with normal saline before its use. According to our previous study [7], the tGCI + HM3379 group received a solution diluted with HM3379 (0.1 mg/kg, *i.p.*) 30 min before and after their surgery, and one dose per day continuously for 14 days. The sham group and the tGCI group received the same amount of normal saline. Neurological deficiency of gerbils was assessed on day 2 and day 14 after tGCI (Figure 1A). Gerbils were anesthetized with phenobarbital sodium (40 mg/kg, *i.p.*), after which blood was collected from their aorta abdominalis. All the animals were sacrificed by decapitation, after which the brain tissues were promptly removed. One part was fixed in 10% formaldehyde for histopathological investigation, and another part immediately put into liquid nitrogen, then stored at −80 °C for protein-level analysis. 

### 2.3. Establishing the tGCI Model

The operating procedure for tGCI was carried out as described by Wang et al. [22]. Briefly, the gerbils were fasted for 12 h but given water freely before surgery and anesthetized with phenobarbital sodium (40 mg/kg, *i.p.*). The bilateral common carotid arteries were exposed and occluded with noninvasive artery clips for 10 min, after which the artery clips were loosened, the blood supply was regained, and the incisions were sutured. The gerbils were returned to their cages for observation. The sham group was operated on as above except that their common carotid artery was not clamped. Surgical instruments, surgical sutures, and suture needles were purchased from Huadong Pharmaceutical Co., Ltd. ( Hangzhou, China). All surgery-related items were washed, sonicated, and sterilized before they were used.

### 2.4. Restraint Stress

Restraint stress was begun on the third day after the tGCI operation and continued for 12 days (Figure 1A) [1]. Gerbils were placed inside a custom-made well-ventilated plastic tube (3 cm in diameter and 10 cm in length) for 4 h per day from 10:00 a.m. to 2:00 p.m. In the tube, they could move their anterior limbs and head but not their body. The sham group was not restricted in any way. After completion of the restraint stress, the gerbils were returned to their cages.

### 2.5. Intracerebroventricular Injection of Short Hairpin RNA (shRNA)-CysLT_2_R or Lentivirus (LV)-CysLT_2_R

The shRNA sequences of CysLT_2_R and the negative control (NC; GenePharma Co. Ltd., Shanghai, China) were as follows (5′-3′): CysLT_2_R, GAT CCC CCC GTC AAC ATG TAT ACT AGC ATT TTC AAG AGA AAT GCT AGT ATA CAT GTT GAC TTT TTG GA A C; and NC, GAT CCC CCC TTC TCC GAA CGT GTC ACG TTT CAA GAG ATT CTC CGA ACG TGT CAC GTT TTT TGG AAC. The cDNA of the double-stranded shRNA oligo was cloned into the pFU-GW-RNAi-GFP lentivirus vector using the Hpa I and Xho restriction enzymes.

The shRNA-CysLT_2_R and LV-CysLT_2_R (Heyuan bio Co., Ltd., Shanghai, China) were administrated by intracerebroventricularly (*i.c.v.*). A stainless steel microinjector, which was placed 0.6 mm posterior to bregma, 1.3 mm lateral from the midline, and 5 mm vertically from the skull surface, was stereotaxically injected into the left lateral ventricle [7]. The shRNA-CysLT_2_R (2 × 10^6^ TU in 4 μL saline) and the LV-CysLT_2_R (1 × 10^9^ TU/mL, 4 ul per site) were injected over 10 min on 2 days and 21 days before tGCI, respectively. The sham group was injected with sterile saline (4 μL) under equivalent conditions. The microinjector was placed for 10 min to reduce backflow, then removed. 

### 2.6. Monitoring the Regional Cerebral Blood Flow (rCBF) 

The gerbils’ heads were shaved and fixed in a flat cranial head position with a stereotaxic instrument (Stoeling, Wood Dale, IL, USA) and sterilized with 75% ethanol. A midsagittal incision was made, and the periosteum was removed with 1% hydrogen peroxide, which exposed the bregma. According to reference [23], the point of origin was set at the bregma, 1.5 mm posterior to the bregma, and 2 mm lateral to the left side of the midline at the marked position and polished with a bone drill at the marked position with gentle movements to prevent penetration of the skull and damage to the dura mater. A fiber optic (Moore, Bridgeport, CT, USA) with a diameter of 0.5 mm was fixed at the marked position with the Loctite 411 instant-drying adhesive (Hobbylinc, Brasellton, GA, USA) and the Insta-Set coagulant (Hobbylinc, Braselton, GA, USA). Then, this fiber optic was connected to the moorVMS-LDF2 laser Doppler flowmeter (Moore, Bridgeport, CT, USA). The changes in the cerebral blood flow in the territory supplied by the middle cerebral artery were monitored in real time with the moorVMS-PC blood-flow recording analysis software (Moore, Bridgeport, CT, USA) and continuously recorded during the tGCI reperfusion (5 min before ischemia, 10 min after clipping, and 10 min of reperfusion). The mean value of the blood flow before ischemia was taken as the baseline value of cerebral blood flow (100%), and the percentage of blood flow at each of the remaining time points was calculated from the flow value/baseline value of 100%. The drop rate of the blood flow during ischemia reached 80% and the following phenomena were simultaneously observed as indications of modeling success: the gerbils were tachypneic and mydriatic, and the cerebral blood flow returned to the basal blood flow within 10 min after the release of the bilateral artery clips, with a regular breathing rate and miosis.

### 2.7. Behavioral Testing

The neurological function of the gerbils was estimated using the modified neurological severity score (mNSS) [24], which assessed the muscle status, abnormal movement, vision, touch, proprioception, and reflex systems of the gerbils. The mNSS test was graded on a numeric scale of 0 to 18 (normal score 0, mild injury score 1–6, moderate injury score 7–12, and severe injury score 13–18). The score was positively correlated with the severity of the neurological injury. The evaluation was carried out by a researcher who was blinded to the experiments on day 2 and day 14 following tGCI. 

A sucrose preference test was conducted to evaluate depression symptoms in the gerbils. Briefly, gerbils received the diet containing 1% (*w*/*v*) sucrose solution, and each gerbil was kept in a separate cage and given two bottles of sucrose solution. After 24 h, one of the bottles was replaced with water for another 24 h. Then, the gerbils were deprived of food and water for 12 h. Afterward, they were again given the sucrose solution and water, and their preference for sucrose was shown by their higher percentage of consumption of the sucrose solution compared to that of the water.

The modified forced swim test was handled as previously described to evaluate despair-like behavior of the gerbils [25]. Before the test, the gerbils were placed into water (at a temperature of 23–25 °C) for 15 min in a glass cylinder that was 20 cm tall and 15 cm in diameter. The water level was deep enough (18 cm) that the tails of the gerbils never touched the bottom. Then, the gerbils were taken out of the water, and put back into their cages. The following day, the gerbils were put through a 6 min forced swim test in a dim environment. The first 2 min were not timed, and the unrelated observers scored the last 4 min of the immobility-floating behavior, which included only the basic movements to keep the head out of the water. These three behavioral tests were detected on day 2 and day 14 after tGCI.

### 2.8. Nissl Staining

The death of neurons was observed in the cerebral cortex via Nissl staining. Briefly, the brain tissues were fixed with formaldehyde and embedded in paraffin, after which 5 μm thick sections were made. The sections were deparaffinized using xylene and dehydrated in a graded series of ethanol. After that, the sections were stained with a 0.2% Nissl staining solution for 5 min. Finally, the changes in the neuronal morphology and density were observed using a light microscope (magnification: 200×). The number of neurons in the cerebral cortex was calculated using the ImageJ software (National Institutes of Health, Bethesda, MA, USA). 

### 2.9. TUNEL Assay

Neuronal apoptosis was detected with a terminal deoxynucleoitidyl transferase dUTP nick end labeling (TUNEL) apoptosis kit (cat# MK1015, Boster Biotech, Wuhan, China) according to the manufacturer’s instructions. The paraffin sections were incubated with TUNEL reaction mixture for 60 min at 37 °C. Double blinding was applied for the quantification. TUNEL-positive cells in three random regions of interest were observed using an inverted fluorescence microscope (magnification: 400×). The apoptotic neurons in the cerebral cortex were analyzed by counting the TUNEL-positive neurons.

### 2.10. Preparation of the Cell Culture 

The mouse microglia cell line BV2 was bought from China Infrastructure of Cell Line Resources (Beijing, China). The BV2 cells were resuspended and cultured in Dulbecco’s modified Eagle’s medium (DMEM; Gibco, Grand Island, New York, NY, USA) added with 10% fetal bovine serum (FBS; Sijiqing bio Co., Ltd., Wuhan, China) and 1% P/S (100 mg/L streptomycin and 100 U/mL penicillin). Then, the cells were incubated in 6-well plates at 4 × 10^5^ cells/well in a normoxic incubator at 37 °C under 5% CO_2_ for 3–4 h to adherent growth [26]. Cells were passaged twice weekly for subsequent analyses.

### 2.11. OGD/Recovery (OGD/R) Model

According to our previous studies [10], the BV2 cells were divided into three groups: the control, OGD/R, and OGD/R + HM3379 groups. HM3379 was diluted to 0.1 μM with the culture medium. The cells were rinsed with PBS twice and added to the Earle’s solution without glucose and incubated in a 37 °C incubator containing a mixture of 95% N_2_ and 5% CO_2_ for 1 h. Following OGD, the cells were cultured with a regular medium in an incubator at 37 °C with 5% CO_2_ to allow recovery in 48 h. The control cells were rinsed and incubated in the Earle’s solution with glucose under normal culture conditions. The other procedures were the same as those for the OGD/R group. 

### 2.12. Analysis of NLRP3 Inflammasome Activation in BV2 Cells

Lipopolysaccharide (LPS, InvivoGen, Carlsbad, CA, USA) stimulation of cells was used as an inflammatory model in vitro. According to Ma et al. [27], nigericin (InvivoGen, Carlsbad, CA, USA), an NLRP3 inflammasome activator, was utilized for investigating the inhibitory efficiency of HM3379 on the activity of NLRP3. The cells in the logarithmic growth phase that were in good condition were cultured at 2 × 10^6^ cells per well in 6-well plates, then exposed to LPS (100 ng/mL) for 4 h. After they were washed with PBS, cells were administrated with HM3379 for 1 h. To activate the NLRP3 inflammasome, the cells were cultured in a serum-free medium that contained nigericin (10 μM). Cells were collected for Western blot analysis.

### 2.13. Quantitative PCR (Q-PCR)

Total RNAs were extracted from the cortical tissue using TRIzol reagent (Invitrogen, Carlsbad, CA, USA) according to previous studies [7,9]. Two μg of RNAs were subjected to reverse transcription using a PrimeScript^TM^RT reagent kit (Takara, Otsu, Japan). CysLT_2_R mRNA expression was detected by Q-PCR using SYBR Premix Ex Taq^TM^ (Takara) on the LightCycler 480 System (Roche Diagnostics, Basel, Switzerland). The primers were designed as follows: CysLT_2_R, forward 5′-TGT CAC CAG TGT CAG GAG TG-3′ and reverse 5′-ACT TTT GAG GAC TCA GCT CCA A-3′; and β-actin, forward 5′-GGC TGT ATT CCC CTC CAT CG-3′ and reverse 5′-CCA GTT GGT AAC AAT GCC ATG T-3′. The results were normalized using β-actin.

### 2.14. Immunofluorescence Staining Analysis

The brain tissues were embedded in paraffin and sectioned into 5 μm thick parts. Then, citrate buffer (pH = 6.0, cat# G1202, Servicebio, Wuhan, China) was utilized for antigen retrieval. The BV2 cells were plated into coverslips in 24-well dishes at 2 × 10^4^ cells/well. Cells were fixed and permeabilized in tubes containing PBS with 0.5% Triton X-100 (cat# P1080, Solarbio, Beijing, China) for 0.5 h. 

The tissue and cell sections were washed with PBS, blocked with 3% BSA for 30 min at room temperature, and then incubated overnight at 4 °C with the following primary antibodies: Iba-1 (a marker of microglia, 1:500, cat# ab178846, abcom, Cambridge, UK), NLRP3 (1:500, cat# NBP2-12446SS, NOVUS, Saint Charles, MO, USA), caspase-1 (1:100, cat# 22915-1-AP, Proteintech, Chicago, IL, USA), and CysLT_2_R (1:500). Following rinsing with PBS, the sections were incubated with a Cy3-conjugated antibody (1:200, cat# A0516, Beyotime, Nantong, China) or an FITC-conjugated secondary antibody (1:500, cat# ab6785, abcam, Cambridge, UK) for 50 min in the dark, followed by Dapi (cat# C1002, Beyotime, Nantong, China) staining for 10 min. The fluorescently stained cells were observed via fluorescence microscopy. 

### 2.15. Western Blot

Proteins were extracted with the cell and tissue total protein extraction kit (cat# KC415, Shanghai Kang Cheng Bioengineering Co., Ltd., Shanghai, China). Briefly, the protein concentration was quantified with a bicinchoninic acid (BCA) protein assay kit (cat# P0010, Beyotime, Nantong, China), and then the protein samples were diluted to the same concentration with a 5 × loading buffer (cat# P0015L, Beyotime) and double-steamed water (cat# A500197-0500, Shanghai Sangon Biotech Co., Ltd., Shanghai, China) and boiled in a metal bath at 100 °C for 10 min. Equal amounts of the proteins were loaded and separated using 10% or 15% SDS-PAGE, and then electro-transferred to polyvinylidene difluoride membranes. The membranes were blocked at room temperature for 1.5 h with 5% non-fat milk or 5% BSA, and then incubated overnight at 4 °C with the following corresponding primary antibodies: NLRP3 (1:1000, cat# 19771-1-AP, Proteintech, Wuhan, China), caspase-1 (1:1000, cat# 22915-1-AP, Proteintech), IL-1β (1:1000, cat# Ab-AF5103, Affinity, Denber, CO, USA), IL-18 (1:1000, cat# ab71495, Abcam), N-domain of GSDMD (GSDMD-N, 1:800, cat# ab219800, Abcam), ASC (1:600, cat# sc-271054, Santa Cruz Biotechnology, Stanta Cruz, CA, USA), and GAPDH (1:1000, cat# 60004-1-Ig, Proteintech). Thereafter, the membranes were incubated with horseradish peroxidase-conjugated goat anti-rabbit IgG (1:3000, cat# GB23303, Servicebio, Wuhan, China) or goat anti-mouse IgG secondary antibodies (1:3000, cat# GB23301, Servicebio) at room temperature for 2 h. The protein bands were visualized with an enhanced chemiluminescence detection kit (Boster Biological Technology, Ltd., Wuhan, China) and a chemiluminescence imaging analyzer. The blot images were processed and quantified with ImageJ software. GAPDH served as an internal control.

### 2.16. Statistical Analysis

The data were analyzed using the Graphpad Prism 8.0 software. All the data were presented as means ± standard errors of the mean. The statistical differences among the groups were verified with one-way ANOVA followed by Newman–Keuls tests. Statistical significance was accepted with *p* < 0.05.

## 3. Results

### 3.1. HM3379 Protected against Depression-like Behaviors and Neurological Injury in PSD Gerbils 

To investigate the anti-depressive effects of HM3379, the tGCI combining restraint stress were performed to produce the PSD model in gerbils. The results of the behavioral tests and the density and morphology of the neurons in the cortical tissues were estimated. The cerebral blood flow obviously decreased under the global cerebral ischemia condition and recovered during the reperfusion in the gerbils, whereas there was no change in the sham group (Figure 1B,C). As expected, the mNSS scores were higher both on day 2 and day 14 in the PSD group than in the sham group. The HM3379 treatment significantly reduced the neurological severity scores in the PSD group (*p* < 0.05, Figure 1D). Furthermore, the percentage of sucrose preference and the immobility time significantly changed between the PSD group and the sham group on day 14 (*p* < 0.05, Figure 1E,F). The HM3379 treatment greatly improved the depression-like behavior in the PSD model on day 14 (*p* < 0.05, Figure 1E,F), although there were no significant differences on day 2 (*p* > 0.05, Figure 1E,F). The Nissl staining showed that the PSD group had fewer Nissl bodies, pyknotic neuronal cell bodies, and deepened staining in the cortex of the ischemia compared with the sham group (1184.00 ± 26.90 vs. 1725.00 ± 32.30, *p* < 0.05; Figure 1G,H). In contrast to the PSD group, HM3379 significantly ameliorated PSD-induced neuronal loss (1441.00 ± 21.71 vs. 1184.00 ± 26.90, *p* < 0.05; Figure 1G,H). In addition, HM3379 treatment diminished the number of TUNEL-positive cells in the cerebral cortex of the PSD group (Figure 2A). These results reveal that HM3379 can ameliorate depression-like behaviors and neurological injury in the PSD model of gerbils.

### 3.2. HM3379 Inhibited Microglial Activation in PSD Gerbils

Next, we explored the effect of HM3379 on PSD-induced microglial activation in the cerebral cortex of the gerbils using immunofluorescence staining. The density of the Iba-1-positive cells, which corresponded to the number of microglia, was higher in the PSD group than in the sham group (461.50 ± 4.16 vs. 403.80 ± 6.16, *p* < 0.05; Figure 2B,C). HM3379 dramatically reduced the number of Iba-1-positive cells compared with the PSD group (436.50 ± 4.20 vs. 461.50 ± 4.16, *p* < 0.05; Figure 2B,C). These data suggest that HM3379 effectively inhibited the activation of microglia under the pathological conditions of PSD in gerbils.

### 3.3. HM3379 Blocked PSD-Induced Activation of the NLRP3 Inflammasome and Pyroptosis in Gerbils

Previous studies have suggested that the activation of the NLRP3 inflammasome and pyroptosis remarkably increased during cerebral I/R injury [20,28]. To elucidate whether the anti-depressive and neuroprotection effects of HM3379 are related to the NLRP3 inflammasome and pyroptosis, the expressions of NLRP3, cleaved caspase-1, mature IL-1β, mature IL-18, GSDMD-N, and ASC in the brain tissues were detected. As shown in Figure 3A,B, the fluorescence intensity of NLRP3 and IL-1β in microglia were obviously upregulated in the PSD group, unlike in the sham group. The number of NLRP3-positive and IL-1β-positive microglia obviously decreased in the PSD group with HM3379 treatment (Figure 3A,B). Furthermore, the Western blot data showed that the expression levels of NLRP3, cleaved caspase-1, mature IL-1β, mature IL-18, GSDMD-N, and ASC were more significantly elevated in the PSD group than in the sham group. Notably, HM3379 significantly downregulated these proteins in the cerebral cortex of the PSD gerbils (NLRP3: 1.48 ± 0.07 vs. 2.40 ± 0.12, *p* < 0.05; cleaved caspase-1: 1.65 ± 0.03 vs. 2.65 ± 0.10, *p* < 0.05; IL-1β: 1.76 ± 0.08 vs. 2.90 ± 0.14, *p* < 0.05; IL-18: 1.60 ± 0.13 vs. 2.82 ± 0.20, *p* < 0.05; GSDMD-N: 1.77 ± 0.08 vs. 3.02 ± 0.06, *p* < 0.05; ASC: 1.48 ± 0.07 vs. 3.01 ± 0.26, *p* < 0.05; Figure 3C,D). These results confirm that HM3379 strongly suppresses the PSD-induced NLRP3 inflammasome/pyroptosis pathway in gerbils.

### 3.4. HM3379 Suppressed OGD/R-Induced NLRP3 Inflammasome Activation and Pyroptosis in BV2 Cells

Following the in vivo results, we established an OGD/R model in the BV2 cells for further study. As shown in Figure 4A, the fluorescence intensity of NLRP3 in the BV2 cells was much stronger in the ODG/R group than in the control group. HM3379 remarkedly reduced the fluorescence intensity of NLRP3 in the BV2 cells. The expressions of NLRP3, cleaved caspase-1, mature IL-1β, mature IL-18, GSDMD-N, and ASC were also more significantly elevated in the OGD/R group than in the control group (*p* < 0.05; Figure 4B,C). Importantly, the increasements of these proteins were dramatically inhibited by HM3379 compared to the OGD/R group (*p* < 0.05; Figure 4B,C). These results showed that HM3379 effectively inhibits OGD/R-induced elevated expressions of the NLRP3 inflammasome and pyroptosis-related proteins in BV2 cells, similarly to the results in vivo.

### 3.5. HM3379 Inhibited Pyroptosis through the NLRP3 Inflammasome Pathway in BV2 Cells

To explore whether the expression of the NLRP3 inflammasome and pyroptosis pathway was determined by the activation of NLRP3 in the BV2 cells, nigericin, as an NLRP3 activator, was used. As shown in Figure 5A,B, expressions of cleaved caspase-1, mature IL-1β, mature IL-18, GSDMD-N, and ASC were dramatically upregulated with nigericin treatment compared to the control group, whereas the HM3379 treatment significantly downregulated these proteins (*p* < 0.05). Taken together, these results indicate that HM3379 could suppress NLRP3 inflammasome activation and subsequent pyroptosis.

### 3.6. HM3379 Protected against PSD-Induced Depression-like Behaviors and Neurological Injury through CysLT_2_R in Gerbils

To further determine whether the anti-depressive and neuroprotection effects of HM3379 were CysLT_2_R-dependent, the shRNA-CysLT_2_R was injected by *i.c.v* 2 days before tGCI to inhibit CysLT_2_R expression, and CysLT_2_R was overexpressed by injection of LV-CysLT_2_R 21 days before tGCI (Figure 6A,B). The white asterisk on the photograph of the TTC-stained coronal slices showed the injection site (Figure 6C). The Q-PCR results showed that, compared with the sham group, the mRNA levels of CysLT_2_R in the PSD, PSD + shRNA-CysLT_2_R, and PSD + LV-CysLT_2_R groups significantly increased. Injection of shRNA-CysLT_2_R or LV-CysLT_2_R more dramatically suppressed or upregulated the mRNA levels of CysLT_2_R than in the PSD group (*p* < 0.05, Figure 6D). The Iba-1 staining indicated that the injection of shRNA-CysLT_2_R reduced the density of the CysLT_2_R-positive microglia, and the injection of LV-CysLT_2_R enhanced the density of the CysLT_2_R-positive microglia with PSD inducement (Figure 6E). The Nissl staining showed that the knockdown CysLT_2_R exhibited a significant increase in neuronal density, and overexpression of CysLT_2_R decreased the neuronal density and the morphological abnormality of the neuronal soma (*p* < 0.05, Figure 6F,G). The HM3379 administration ameliorated the neuronal injury in the PSD + LV-CysLT_2_R group but did not change the PSD + shRNA-CysLT_2_R group (Figure 6F,G). Importantly, the depression-like behaviors also improved with HM3379 in the PSD + LV-CysLT_2_R group, and there were no changes in the PSD + shRNA-CysLT_2_R group (Figure 6H,I). These data indicate that CysLT_2_R is involved in neuron damage in the PSD model, and HM3379 exhibits anti-depressive and neuroprotection effects depending on CysLT_2_R.

## 4. Discussion

Several major new findings arose from our study. We have provided evidence that HM3379 contributed to the improvement of chronic neuronal injury and depression-like behaviors, and suppressed microglial activation in PSD gerbils. Importantly, HM3379 inhibited protein expressions of the NLRP3 inflammasome, as well as pyroptosis, both in the PSD gerbils and the BV2 cells under the OGD/R condition. Furthermore, we demonstrated that HM3379 suppressed pyroptosis through mediating NLRP3 inflammasome activation in BV2 cells. HM3379 protected against PSD-induced depression-like behaviors depending on CysLT_2_R in vivo. Taken together, the beneficial effects of HM3379 on PSD-induced depressive-like behaviors and chronic neuroinflammation were partly via the NLRP3 inflammasome/pyroptosis pathway in gerbils (Figure 7).

It is well-known that Mongolian gerbils more typically develop GCI models than rats and mice because of the variable absence of the Willis circle from its basilar cerebral artery [24,29]. Thus, we used Mongolian gerbils subjected to tGCI and spatial restraint stress as an experimental model of PSD. Recent research has demonstrated that microglia can be activated in cerebral ischemia insult, and the activated microglia further releases proinflammatory cytokines and exhibits an ability to enhance phagocytic activity [14,30,31]. In this study, we have revealed that, in both the PSD model and the OGD/R-induced BV2 cell model, microglia were activated, and their persistent activation promoted the producing of inflammatory cytokines, including IL-1β and IL-18, which corroborates findings from previous studies. Therefore, we attempted to find potential therapeutic strategies that could ameliorate chronic neuroinflammation after PSD.

Clinical studies have indicated that dysfunction or repair is lasting in patients with ischemic stroke, and the protective effects of drugs rarely exceed 7 days after ischemia [32,33,34]. Thus, the long-term neuroprotective effect of drugs is extremely important in evaluating the clinical condition of patients. HM3379, as a selective CysLT_2_R antagonist, is known to protect against cerebral ischemia injury [10]. A previous study elucidated that CysLT_2_R inhibition was accompanied with attenuation of microglia-related neuroinflammation in ischemic stroke [13]. Our previous study determined that 0.1 μM HM3379 was the most effective dose for the improvement of neuronal damage by intraperitoneal injection [7]. In this study, we confirmed that HM3379 administration ameliorated neuronal loss, improved the mNSS score and depressive-like behavior, and inhibited microglial activation in PSD gerbils. This efficiency was CysLT_2_R-dependent. Nevertheless, the mechanisms that underlined neuroprotection in PSD gerbils were not well discussed.

Recently, the NLRP3 inflammasome, as the best characterized inflammasome, was extensively expressed in the microglia of the ischemic brain tissue [15,18,35]. Oligomerized NLRP3 recruits the adaptor molecule ASC through PYD–PYD interactions to form assembled ASC, which facilitates the recruitment of procaspase-1 through CARD–CARD interactions to generate the NLRP3 inflammasome complex [36]. The activation of NLRP3 inflammasome causes the self-cleavage of procaspase-1, followed by the inducement of IL-1β and IL-18 maturation and secretion [19,37]. Abundant evidence has reported that the NLPR3 inflammasome was activated in the ischemic stroke model and suppressed the activity of the model, which led to amelioration of the neurological deficit scores [38,39]. Our study showed that HM3379 downregulated the expressions of NLRP3, cleaved caspase-1, and mature IL-1β/IL-18 during in vivo and in vitro studies. Notably, we also found that the pharmacological activity of NLRP3 with nigericin, an NLRP3 activator, obviously elevated the expressions of NLRP3 inflammasomes, such as of cleaved caspase-1 and mature IL-1β/IL-18, while HM3379 diminished these increases in the LPS-induced BV2 cells. Our results indicate that HM3379 may ameliorate PSD-induced chronic neuroinflammation partly via inhibiting the NLRP3 inflammasome pathway in gerbils.

Pyroptosis, a novel programmed cell death pathway, is initiated by the activation of inflammatory caspases [40]. It has several distinct qualities over other types of cell death, including membrane pore formation, cell swelling, and membrane rupture, which result in leakage of the proinflammatory cytosolic contents [17,18,19]. Previous studies have shown that pyroptosis plays a crucial role in the development of tissue damage [41]. It has been noted that GSDMD, which is located downstream of the inflammatory caspase, is a 53 kDa executive protein for pyroptosis [42]. GSDMD-N forms pores on the intracellular membrane and causes pyroptosis through the destruction of the membrane [43,44]. In this study, we observed that HM3379 could inhibit the upregulated expression of ASC and GSDMD-N both in the PSD and OGD/R models. Furthermore, the expression levels of ASC and GSDMD-N were also elevated after NLRP3 activation, and HM3379 significantly reversed such elevations in the BV2 cells.

This study had a limitation. The efficiency of HM3379 action on depression beyond tGCI was not confirmed in the gerbils. However, the phenotypes observed in the PSD model were indeed significant, and the mechanism of the NLRP3 inflammasome/pyroptosis pathway was intriguing. Future experiments need to be authorized on a depression model at the individual level.

## 5. Conclusions

In summary, our present study confirmed that the CysLT_2_R antagonist HM3379 diminished PSD-induced neurological injury and depression-like behaviors in gerbils, which was possibly related to the suppression of the NLRP3 inflammasome/pyroptosis pathway. Consequently, the promising effect of HM3379 on the amelioration of PSD-induced chronic neuroinflammation may present a considerable clinical advantage for PSD therapy.

## Figures and Tables

**Figure 1 brainsci-12-00976-f001:**
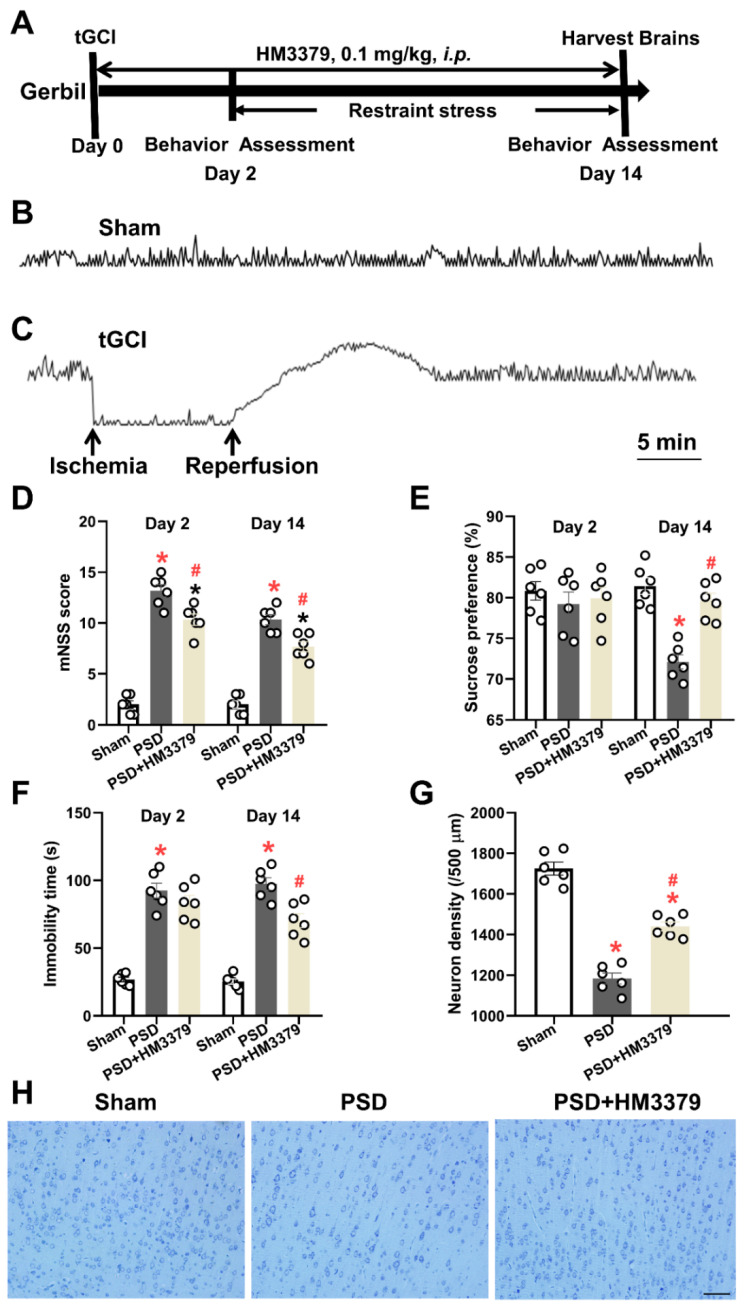
HM3379-affected depression-like behaviors and neurological injury in the PSD gerbils. (**A**) Schematic diagram of the experimental protocol. The gerbils were subjected to tGCI, and restraint stress was initiated on day 2 after tGCI surgery for 12 consecutive days to make a PSD model. HM3379 (0.1 mg/kg, *i.p.*) was administrated for 14 days. The neurobehavioral outcomes and depressive-like behavior were assessed on the 2nd and 14th days. Typical recordings show rCBF in gerbils with sham operation (**B**) and tGCI-reperfusion surgery (**C**). (**D**) HM3379 post-conditioning showed a significant decrease in mNSS at day 2 and day 14 post-injury. HM3379 improved depression-like behavior in sucrose preference test (**E**) and forced swim test (**F**) in PSD gerbils. (**G**) Neuronal density in the cortex regions. Neuronal loss in PSD gerbil model was significantly ameliorated by HM3379. (**H**) Representative photomicrographs of Nissl-stained brain sections. Data are presented as mean ± SEM. * indicates *p* < 0.05 vs. Sham, ^#^ indicates *p* < 0.05 vs. PSD. N = 6 per group. Scale bars = 100 µm. PSD, post-stroke depression. tGCI, transient global cerebral ischemia. rCBF, regional cerebral blood flow. mNSS, modified neurological severity score.

**Figure 2 brainsci-12-00976-f002:**
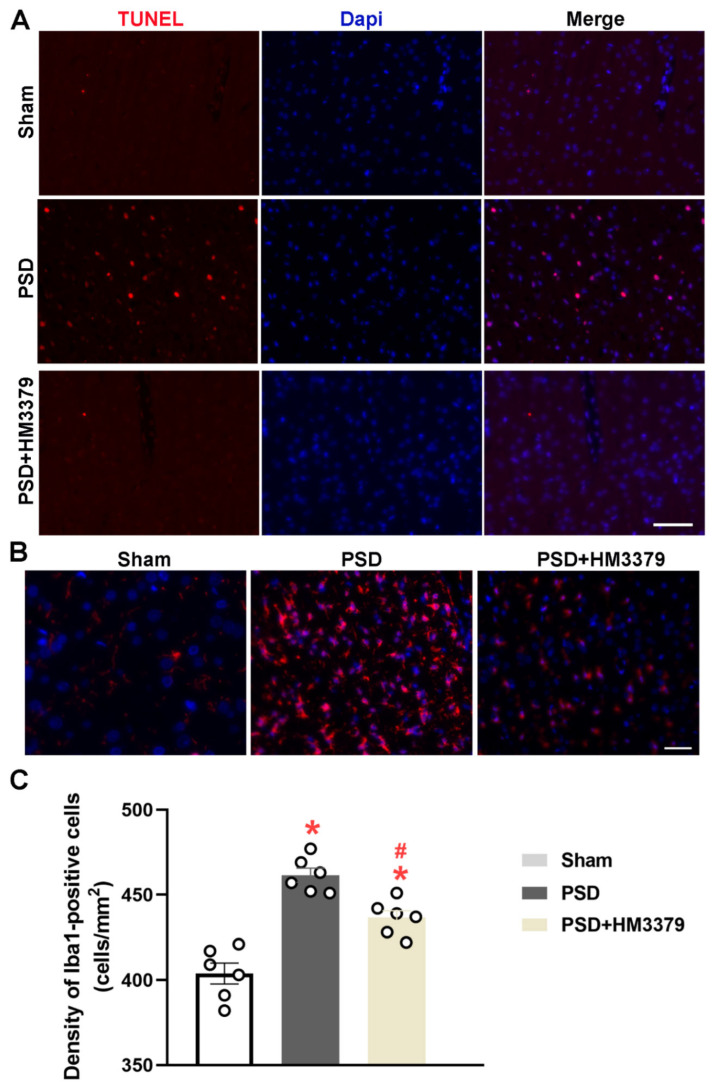
HM3379 attenuated neurocyte apoptosis and inhibited microglial activation in the PSD gerbils. (**A**) TUNEL (red) staining was localized in the nuclei of neurons as shown in the PSD group. Fewer TUNEL-positive cells were observed in the PSD + HM3379 group than the PSD group. N = 3 per group. Scale bars = 100 µm. (**B**) Representative immunofluorescent staining of the Iba-1-positive cells in the cortex of PSD gerbils. (**C**) The density of the Iba-1-positive cells was quantified. Data are expressed as mean ± SEM. * indicates *p* < 0.05 vs. sham, ^#^ indicates *p* < 0.05 vs. PSD. N = 6 per group. Scale bar = 100 µm. PSD, post-stroke depression. TUNEL, transferase dUTP nick end labeling.

**Figure 3 brainsci-12-00976-f003:**
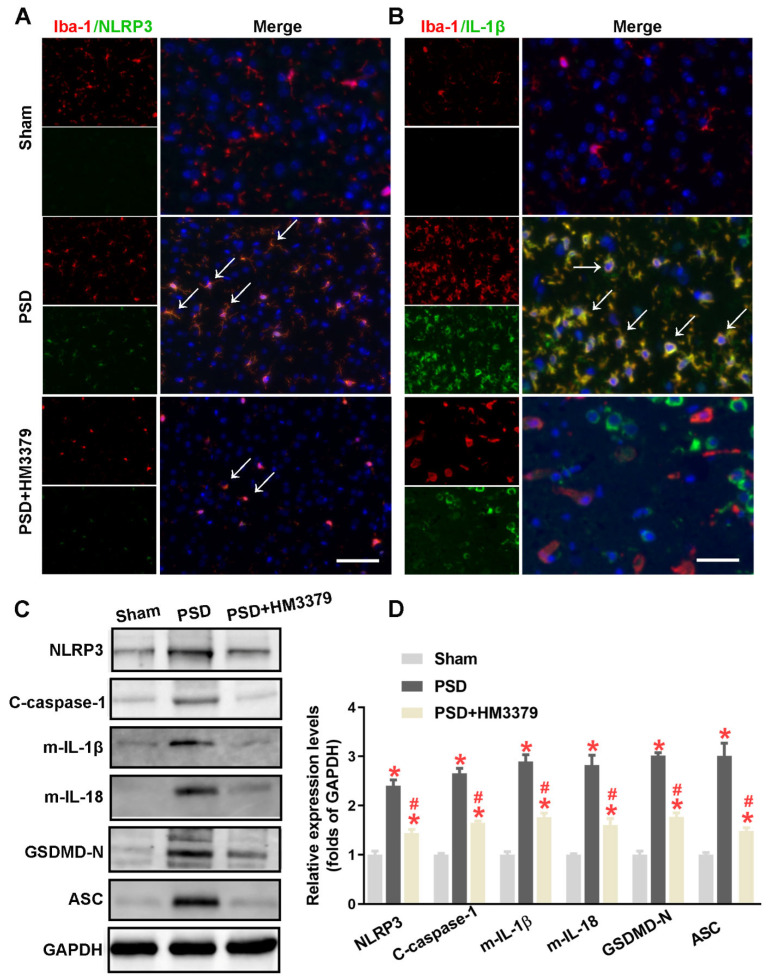
HM3379 blocked PSD-induced activation of the NLRP3 inflammasome and pyroptosis in gerbils. (**A**) Immunofluorescence staining of Iba-1 (red) and NLRP3 (green) revealed NLRP3 expression in microglia. (**B**) Brain sections were stained with IL-1β (green), as well as Iba-1 (red) or Dapi (blue), to monitor IL-1β accumulation. White arrows point to NLRP3-positive or IL-1β-positive in microglia on day 14 after tGCI. (**C**) Western blot analysis of the expression of NLRP3, cleaved caspase-1, mature IL-1β, mature IL-18, GSDMD-N, and ASC in sham, PSD, and PSD + HM3379 group. (**D**) Combined Western blot statistics. Data are expressed as mean ± SEM. * indicates *p* < 0.05 vs. sham, ^#^ indicates *p* < 0.05 vs. PSD. N = 3 per group. Scale bars = 50 μm. PSD, post-stroke depression. tGCI, transient global cerebral ischemia.

**Figure 4 brainsci-12-00976-f004:**
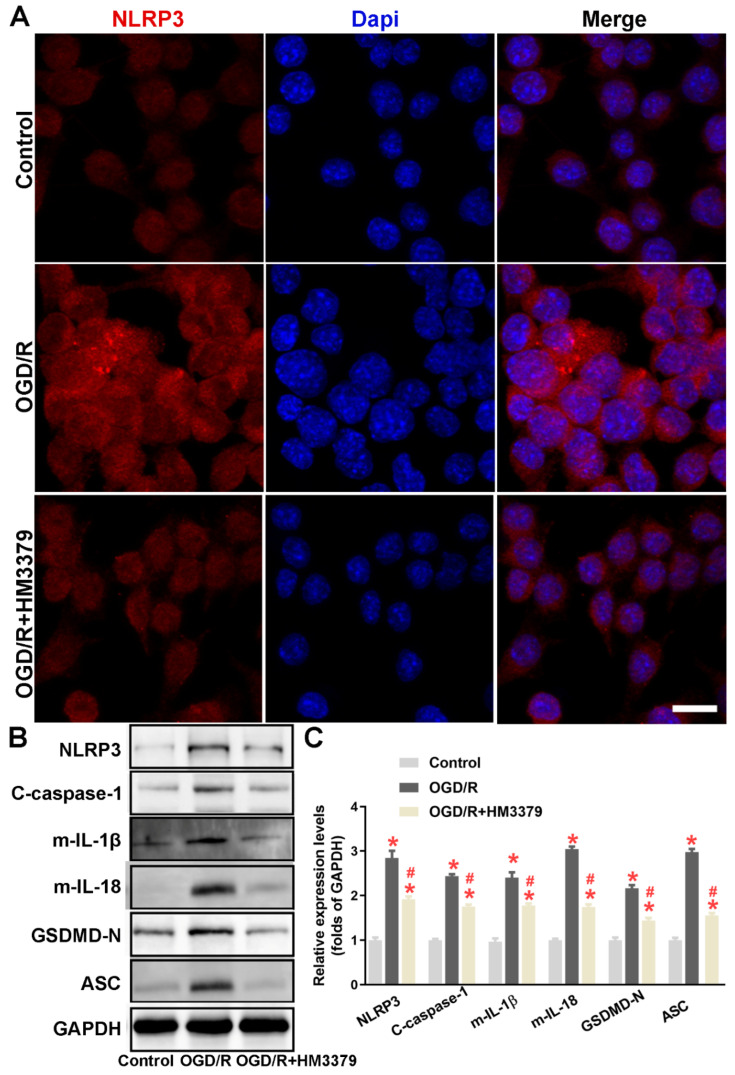
H3379 suppressed OGD/R-induced NLRP3 inflammasome activation and pyroptosis in BV2 cells. (**A**) NLRP3 (red) immunofluorescence staining results showed that HM3379 decreased the intensity of NLRP3-positive cells when exposed to the OGD/R condition. (**B**) The expressions of NLRP3, cleaved caspase-1, mature IL-1β, mature IL-18, GSDMD-N, and ASC in the indicated groups were determined by Western blot. (**C**) Combined Western blot statistics. Data are presented as mean ± SEM. * indicates *p* < 0.05 vs. control, ^#^ indicates *p* < 0.05 vs. OGD/R. N = 3 per group. Scale bars = 50 μm. OGD/R, oxygen glucose deprivation/ reperfusion.

**Figure 5 brainsci-12-00976-f005:**
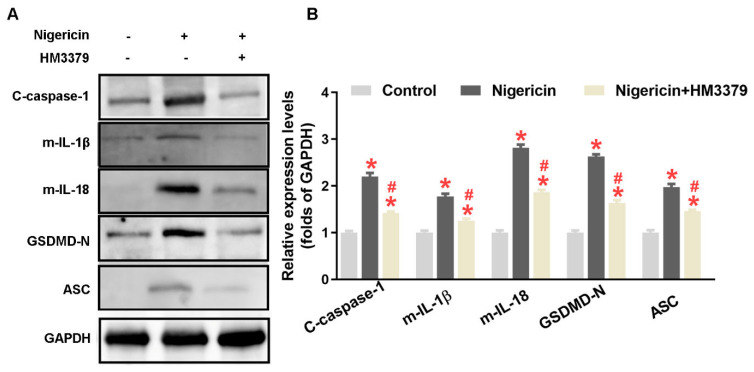
HM3379 inhibited pyroptosis through NLRP3 inflammasome pathway in BV2 cells. (**A**) The expression levels of NLRP3 inflammasome and pyroptosis-related proteins, such as cleaved caspase-1, mature IL-1β, mature IL-18, GSDMD-N, and ASC. (**B**) The effect of nigericin, a NLPR3 activator, on the above protein expressions in BV2 cells stimulated with LPS and followed by HM3379 treatment. Data are presented as mean ± SEM. * indicates *p* < 0.05 vs. control, ^#^ indicates *p* < 0.05 vs. nigericin. N = 3 per group. LPS, lipopolysaccharide.

**Figure 6 brainsci-12-00976-f006:**
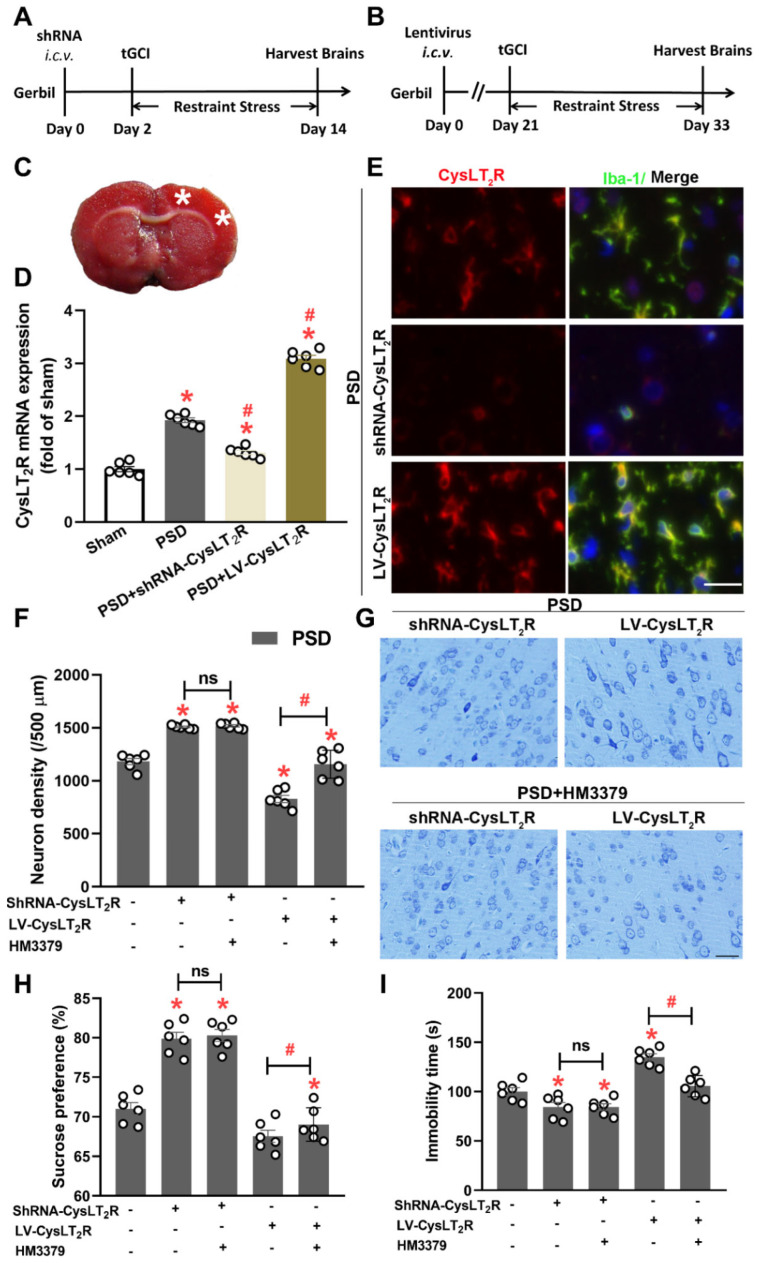
HM3379 protected against PSD-induced depression-like behaviors through CysLT_2_R in gerbils. (**A**,**B**) Schematic diagram of injected shRNA-CysLT_2_R and LV-CysLT_2_R by *i.c.v*. (**C**) The white asterisk on the photograph of TTC-stained coronal slices showed the injection site. (**D**) Quantitative analysis of CysLT_2_R mRNA expression in sham, PSD, PSD + shRNA-CysLT_2_R, and PSD + LV-CysLT_2_R groups. (**E**) Immunofluorescence staining of Iba-1 (green) and CysLT_2_R (red) revealed CysLT_2_R expression in microglia after injection of shRNA-CysLT_2_R and LV-CysLT_2_R. (**F**) Neuronal density in the cortex regions. HM3379 significantly ameliorated neuronal loss in the PSD-induced gerbil model by inhibiting CysLT_2_R. (**G**) Representative photomicrographs of Nissl-stained brain sections. HM3379 depending on CysLT_2_R improved depression-like behavior in sucrose preference test (**H**) and forced swim test (**I**) in PSD gerbils. Data are presented as mean ± SEM. * indicates *p* < 0.05 vs. sham, ^#^ indicates *p* < 0.05 vs. PSD. N = 6 per group. Scale bars = 50 µm. ns, no significant difference. PSD, post-stroke depression. tGCI, transient global cerebral ischemia.

**Figure 7 brainsci-12-00976-f007:**
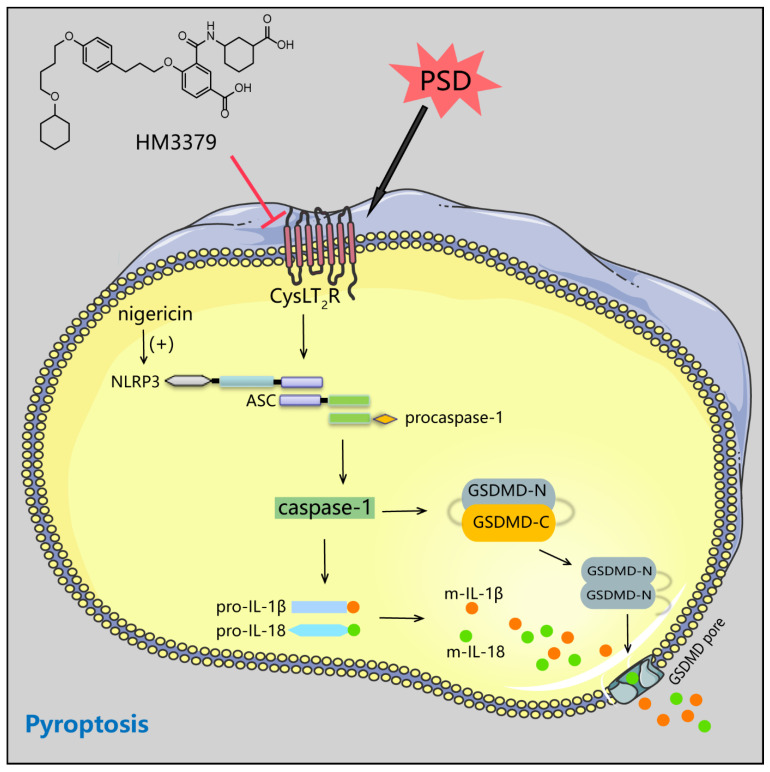
Schematic diagram for of HM3379 inhibition of the NLRP3 inflammasome/pyroptosis pathway ameliorating post-stroke depression in gerbils.

## Data Availability

The data presented in this study are available on request from the corresponding authors.

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
