# Peer review of "CysLT2R Antagonist HAMI 3379 Ameliorates Post-Stroke Depression through NLRP3 Inflammasome/Pyroptosis Pathway in Gerbils"

_brainsci, 2022, doi:10.3390/brainsci12080976_

Round 1

Reviewer 1 Report

The work was conducted to investigate the mechanistic of HM3379 in ameliorating post-stroke depression symptoms and neuroinflammation. The experiments were well-designed to test hypotheses with clear outcomes. Overall, the manuscript was written in good English but some improvements need to be made in the grammar and spelling. The methods section has good clarity but additional details can be added which would be helpful to readers. The data are well-presented but in some cases, the labels for statistical significance are too small to make out the difference between '*' and '#'. Below are specific comments for further improvement of the manuscript:

Methods

1. For monitoring of regional CBF, it is stated that this was conducted 5 min before ischemia, 10 min after clipping, and 10 min reperfusion. Was it after clipping or during clipping? Was the monitoring conducted in parallel to the tGCI surgery?

2. When were sucrose preference and forced swimming tests conducted? Are these tests part of the behavior assessment conducted on Day 2 and Day 4?

3. Which samples were used for the TUNEL assay? Were they part of the brains which were cryopreserved?

4. Suggest adding the details on why 0.1 microM was optimal i.e. based on the previous study, what was found when 0.1 microM HM3379 was used.

Results

1. Please correct Figure 1 legend - HM3379.

2. Quality of the figures could be improved in particular Figure 1. It is too small and difficult to differentiate * and #.

3. Figure 5A – Please edit HAMI3379 to HM3379 for consistency.

Spelling mistake line 479 - Pervious.

Author Response

Reviewer #1:

Synopsis: The work was conducted to investigate the mechanistic of HM3379 in ameliorating post-stroke depression symptoms and neuroinflammation. The experiments were well-designed to test hypotheses with clear outcomes. Overall, the manuscript was written in good English but some improvements need to be made in the grammar and spelling. The methods section has good clarity but additional details can be added which would be helpful to readers. The data are well-presented but in some cases, the labels for statistical significance are too small to make out the difference between '*' and '#'. Below are specific comments for further improvement of the manuscript:

Comment #1-1: For monitoring of regional CBF, it is stated that this was conducted 5 min before ischemia, 10 min after clipping, and 10 min reperfusion. Was it after clipping or during clipping? Was the monitoring conducted in parallel to the tGCI surgery?

Response: Thank you very much for your kind suggestion. The monitoring of regional CBF was during clipping, and monitoring conducted in parallel to the tGCI surgery. This information has been showed in the revised manuscript.

Comment #1-2: When were sucrose preference and forced swimming tests conducted? Are these tests part of the behavior assessment conducted on Day 2 and Day 4?

Response: Thank you very much for your kind suggestion. The sucrose preference and forced swimming tests were conducted on day 2 and day 14 after tGCI. There were no significant differences of sucrose preference between the PSD group and the sham group on day 2 (P > 0.05, Figure 1E). In addition, there were no significant differences of immobility time between the PSD group and the PSD+HM3379 group on day 2 (P > 0.05, Figure 1F). The information has been added to the revised manuscript.

Comment #1-3: Which samples were used for the TUNEL assay? Were they part of the brains which were cryopreserved?

Response: Thank you very much for your kind suggestion. The cerebral cortex in all groups was used for TUNEL assay. Brain tissues were fixed with formaldehyde, embedded in paraffin, and then made 5 μm-thick sections. Paraffin sections were then stained with TUNEL kit. The content was provided in the revised manuscript.

Comment #1-4: Suggest adding the details on why 0.1 microM was optimal i.e. based on the previous study, what was found when 0.1 microM HM3379 was used.

Response: Thank you very much for your kind suggestion. Our previous study has determined that 0.1 μM HM3379 was the most effective dose for improvement of neuronal damage by intraperitoneal injection in MCAO rat model (Neuroscience, 2015, 291: 53-69). The content has been provided in the revised manuscript.

Comment #1-5: Please correct Figure 1 legend - HM3379.

Response: Thank you very much for your kind suggestion. The legend of Figure 1 has been changed to HM3379 in the revised manuscript.

Comment #1-6: Quality of the figures could be improved in particular Figure 1. It is too small and difficult to differentiate * and #.

Response: Thank you very much for your kind suggestion. The signs in all figures have been enlarged and marked into red color.

Comment #1-7: Figure 5A – Please edit HAMI3379 to HM3379 for consistency.

Response: Thank you very much. HAMI3379 in Figure 5A has been changed into HM3379 in the revised manuscript.

Comment #1-8: Spelling mistake line 479 - Pervious.

Response: Thanks. The word “Pervious” has been corrected in the revised manuscript.

Reviewer 2 Report

This work on post-stroke depression and its treatment with an anti-inflammatory drug is potentially interesting. However, I remark that the experimental design is incomplete and thus cannot provide robust evidence for the authors’ hypothesis. Specifically, authors considered the following factors: (i) the ischemic lesion; (ii) exposure to stress to induce depression; (iii) drug treatment to counteract depression. This is indeed a factorial design in which the appropriate experimental groups should be eight (2x2x2): healthy and damaged animals all exposed to stress (restraint) or not, receiving the drug or its vehicle. In such a case, all the required control groups could be considered to correctly evaluate the experimental outcomes. Other minor issues:

-          English requires correction and/or rewriting.

-          Add a reference for the stress model (e.g., DOI10.1023/A:1022969828885).

-          Figures are too small.

Author Response

Reviewer #2:

Synopsis: This work on post-stroke depression and its treatment with an anti-inflammatory drug is potentially interesting. However, I remark that the experimental design is incomplete and thus cannot provide robust evidence for the authors’ hypothesis. Specifically, authors considered the following factors: (i) the ischemic lesion; (ii) exposure to stress to induce depression; (iii) drug treatment to counteract depression. This is indeed a factorial design in which the appropriate experimental groups should be eight (2x2x2): healthy and damaged animals all exposed to stress (restraint) or not, receiving the drug or its vehicle. In such a case, all the required control groups could be considered to correctly evaluate the experimental outcomes.

Response: Thank you very much for your kind suggestion. Our previous study has demonstrated that HM3379 significantly decreased cerebral infarct volume and improved neurological damage in middle cerebral artery occlusion-induced rat model (Brain research, 2012, 1484: 57-67). This study suggests therapeutic potential for CysLT2R antagonists HM3379 in the treatment of ischemic stroke. In the current study, we focus to investigate the role of HM3379 in post-stroke depression induced chronic neuroinflammation and related mechanisms in gerbils. Our results demonstrate that HM3379 exhibits beneficial effects on PSD, which are partially through suppressing NLRP3 inflammasome/pyroptosis pathway. There is no study has been reported the anti-depression effect of CysLT2R or HM3379 in rodent. Once our results are confirmed, we will conduct experiments on depression-like model in future studies. We also discussed it as a limitation in the revised manuscript.

Other minor issues:

Comment #2-1: English requires correction and/or rewriting.

Response: Thank you very much for your kind suggestion. The English writing of our manuscript has been reviewed by a native speaker. We have marked all changes clearly in the revised manuscript.

Comment #2-2: Add a reference for the stress model (e.g., DOI10.1023/A:1022969828885).

Response: Thank you very much for your kind suggestion. We did not find the reference “DOI10.1023/A:1022969828885” on the pubmed website. We have provided a reference for the stress model (Int J Mol Sci, 2020, 21(10):3454).

Comment #2-3: Figures are too small.

Response: Thank you very much for your kind suggestion. The signs in all figures have been enlarged and marked into red color.

Reviewer 3 Report

dear authors, with interest i read your papee which is well writen and explained, however i found some very old references(8,23,29,32,33)

also, you did not mentioned the study limitation,please write you study's limitations 1ns update your old reference

Author Response

Reviewer #3:

Comment #3-1: dear authors, with interest I read your paper which is well wrote and explained, however I found some very old references (8,23,29,32,33)

Response: Thank you very much for your kind suggestion. The old references (23, 29, 31, 32) have been updated in the revised manuscript. The reference 8 was our previous study which suggests therapeutic potential for HM3379 in the treatment of ischemic stroke (Brain research, 2012, 1484: 57-67). The current study was studied based on reference 8, so it was irreplaceable.

Comment #3-2: also, you did not mention the study limitation, please write you study's limitations 1ns update your old reference

Response: Thank you very much for your kind suggestion. The limitations have been provided in the “Discussion” section of the revised manuscript.

Round 2

Reviewer 2 Report

The experimental design of the study was not modified as recommended. This means that the statistical evaluation of the power was deeply affected by the lack of an appropriate experimental design so to prevent the appropriate calculation of the sample size.